# The Study of Cerebrospinal Fluid microRNAs in Spinal Cord Injury and Neurodegenerative Diseases: Methodological Problems and Possible Solutions

**DOI:** 10.3390/ijms23010114

**Published:** 2021-12-22

**Authors:** Irina Baichurina, Victor Valiullin, Victoria James, Albert Rizvanov, Yana Mukhamedshina

**Affiliations:** 1Omics Technologies Research Laboratory, Institute of Fundamental Medicine and Biology, Kazan Federal University, 420008 Kazan, Russia; 2Department of Histology, Cytology, and Embryology, Kazan State Medical University, 420012 Kazan, Russia; valiullinvv@yandex.ru (V.V.); yana.k-z-n@mail.ru (Y.M.); 3Division of Biomedical Science, School of Veterinary Medicine and Science, Faculty of Medicine and Health Sciences, University of Nottingham Biodiscovery Institute, University Park, Nottingham NG7 2RD, UK; victoria.james@nottingham.ac.uk; 4Clinical Research Center for Precision and Regenerative Medicine, Institute of Fundamental Medicine and Biology, Kazan Federal University, 420008 Kazan, Russia; rizvanov@gmail.com

**Keywords:** neurological disorders, spinal cord injury, neurodegenerative diseases, miRNA, cerebrospinal fluid, biomarkers

## Abstract

Despite extensive research on neurological disorders, unanswered questions remain regarding the molecular mechanisms underpinning the course of these diseases, and the search continues for effective biomarkers for early diagnosis, prognosis, or therapeutic intervention. These questions are especially acute in the study of spinal cord injury (SCI) and neurodegenerative diseases. It is believed that the changes in gene expression associated with processes triggered by neurological disorders are the result of post-transcriptional gene regulation. microRNAs (miRNAs) are key regulators of post-transcriptional gene expression and, as such, are often looked to in the search for effective biomarkers. We propose that cerebrospinal fluid (CSF) is potentially a source of biomarkers since it is in direct contact with the central nervous system and therefore may contain biomarkers indicating neurodegeneration or damage to the brain and spinal cord. However, since the abundance of miRNAs in CSF is low, their isolation and detection is technically difficult. In this review, we evaluate the findings of recent studies of CSF miRNAs as biomarkers of spinal cord injury (SCI) and neurodegenerative diseases. We also summarize the current knowledge concerning the methods of studying miRNA in CSF, including RNA isolation and normalization of the data, highlighting the caveats of these approaches and possible solutions.

## 1. Introduction

Neurological disorders include a wide range of diseases that affect the brain, spinal cord and nerves, leading to various symptomatic manifestations. For most neurological diseases, the effectiveness and outcome of treatment depends on early diagnosis. Currently, diagnosis, especially in the case of neurodegenerative diseases, is based on detection of clinical symptoms and detection of structural changes by neuroimaging. However, manifestation of the first clinical symptoms often occurs in the later stages of the disease, when pathological processes have led to irreversible molecular and cellular shifts. These molecular changes that occur during the course of the disease could be tracked much earlier [1,2]. In this regard, studies that search for early biomarkers of the pre-symptomatic stages of neurological disorders are highly relevant, not only for diagnostics but also in the staging of disease and choice of therapeutic intervention [3].

Spinal cord injuries (SCI), which lead to impaired sensory, motor and autonomic functions that affect the physical, psychological and social state of the patient, are one example of where effective molecular biomarkers would have a marked clinical benefit [4,5]. The number of cases of SCI is growing, often associated with car accidents and various types of falls [6]. The neuroregenerative potential of the central nervous system (CNS) is rather low; nevertheless, some patients with SCI manage to fully or partially restore lost functions. [7,8]. Currently, clinical manifestations of SCI are assessed and classified using functional neurological examination, stratifying injury severity using the American Spinal Injury Association (ASIA) Impairment Scale (AIS). The reliability of the developed scale is well-tested, but nevertheless, the variability of spontaneous recovery within each AIS grade is high [9]. Establishing a panel of biomarkers that would objectively classify the severity of SCI and have a prognostic value would enable the optimal therapeutic strategy and better management of each patient’s condition [10,11].

With the development of high-throughput methods of genomic and transcriptome analysis, researchers have sorted genes, the expression change of which triggers inflammatory and degenerative processes in SCI [12,13]. Among the known post-transcriptional regulators that affect gene expression, miRNAs are considered to be important factors, as they are able to regulate more than half of all genes in the human genome [14]. These endogenous, small, non-coding RNAs of 18–24 nucleotides in length [15] have been shown to change under different physiological and pathological conditions and can be detected in biological fluids [16]. miRNAs have already been identified as being involved in the development of several neurological disorders, including central nervous system (CNS) injuries and neurodegenerative diseases [12,17].

Taken together, the data from recent studies strongly support miRNAs as potential new targets for the detection and regulation of post-traumatic and neurodegenerative processes [15,18,19]. Changes in miRNA expression in the abovementioned disorders can be detected in CSF, serum and plasma since miRNAs enclosed in exosomes are able to cross the blood-brain barrier [20]. However, the miRNA profile in CSF is also due to the direct contact with the central nervous system and, as a result, may provide a more objective reflection of molecular changes than plasma or serum. Due to the technical difficulties and cost of studying miRNA profiles in CSF, this approach is not widely implemented. In this regard, the purpose of this review is to systematize the guidelines for studying the miRNA profile of CSF in SCI and other neurological disorders in order to reveal existing problems and potential solutions.

## 2. Detection of miRNAs in Cerebrospinal Fluid of Patients with Neurological Disorders

miRNAs are stable in body fluids and can be useful as non-invasive biomarkers in disease diagnosis and prognosis [3,16,21,22]. Circulating extracellular miRNAs can be contained within vesicles, such as exosomes, as well as bound within proteins and lipoprotein complexes, which makes them resistant to RNases within the external environment [23]. However, these circulating extracellular miRNAs are often in low abundance, making their detection in small volumes of biofluids technically challenging [24,25,26]. There are several examples of the involvement of miRNAs during SCI and neurodegenerative diseases, which will be discussed below, alongside the methodological difficulties in isolating these miRNAs from CSF and challenges in normalizing expression of these circulating miRNAs for clinical use.

### 2.1. Spinal Cord Injury (SCI)

There is a hypothesis that expression of miRNA in the spinal cord is specific and persists during the evolution of vertebrates [12]. Experimental data show that some miRNAs are cell-specific, for example, miR-124 and miR-128, which are predominantly expressed in neurons. miRNAs are involved in cellular changes in the damaged central nervous system, where they play an active role in the regulation of inflammation, apoptosis, cell proliferation and differentiation [27].

Early studies of miRNA profiles were carried out in the injured tissue of the spinal cord of rats and mice during the acute period of injury [12,17,28,29,30,31,32]. At days 1 and 7 after SCI, suppression of seven miRNAs was observed, the potential targets of which are tumor necrosis factor α (TNF-α), interleukin 1β (IL-1β), and intercellular adhesion molecules 1 (ICAM1), indicating loss of these miRNAs leads to activation of inflammatory processes [17]. The authors found that two of the seven miRNAs (miR-181a and miR-127) can affect the expression level of cytosolic and secretory phospholipases A2 (cPLA 2 and sPLA 2) (Appendix A), the activation of which plays an important role in the pathogenesis of secondary damage in SCI [17,33]. In a similar period of time after SCI, increased expression of miR-221 and miR-1 was also observed, which target anti-inflammatory genes *annexin A1* and *A2* [17]. MiR-221 has also been linked with neurodegeneration by acting on EGFR [34]. Similar increases in the expression of miR-1, miR-206, miR-152 and miR-214 were also reported, affecting expression of antioxidant genes (*SOD1* and *catalase*) [17].

Strickland et al. (2011), used a similar animal model of SCI to study the level of miRNA expression in the tissue of the injured spinal cord in both the acute phase (1 and 4 days) and the subacute period (14 days) of neurotrauma (Appendix A). The expression of two miRNAs (activation of miR-146a and suppression of miR-129-2) were found to change significantly after SCI compared to a sham control. The initial severity of SCI was found to be inversely related to later expression of miR-146a and miR-129-2. Increased expression of these miRNAs was observed from 4 to 14 days in less severe injuries. Regression analysis showed that 74.6% of changes in miR-129-2 expression on days 4 and 14 were explained by variations in baseline assessments of hindlimb motor activity, assessed by the BBB scale. On the other hand, 69.7% of the variation in miR-146a expression on day 14 was attributed to variations in the same parameter 24 h after SCI. Thus, it was concluded that expression of miR-129-2 and miR-146a correlates well with functional assessment after injury [28].

A year later, Yunta et al. (2012) conducted a similar study to analyze the miRNA profile in the tissue of the injured spinal cords of rats on days 1, 3 and 7 of neurotrauma. As in the work of Strickland et al. (2011), they noted a statistically significant increase in miR-146a expression at 7 days post injury (dpi) in comparison to the control groups. The authors showed that 7 days after SCI, 59 out of 187 genes activated after moderate and mild injuries were targets for these upregulated miRNAs; these genes demonstrated significant changes in expression as compared to the intact control. At the same time, significant decreases in the level of 25 miRNAs led to increased expression of genes involved in key processes in the pathophysiology of SCI (cell death and inflammation) on the 7th day after SCI [12].

The study of miRNAs in SCI is not limited to the area of damage alone [25,26]. The work of Tigchelaar et al. (2017) investigated the expression of miRNAs in serum and CSF of large animals (pigs) on days 1, 3 and 5 after contusion SCI of varying severity. The study found a decrease in two serum miRNAs, miR-1285 and miR-4331, and an increased content of serum miR-133b, miR-1, miR-885-5p, miR-204 and miR-208b after SCI in pigs. The authors also obtained changes in the total abundance of miRNA on days 1 and 3 after SCI of any severity, which significantly correlated with structural and functional parameters. These results suggest that total serum miRNA levels are associated with severity of injury and therefore may have predictive value for recovery from SCI [35].

Tigchelaar et al. (2019) extended their work to investigate the profile of miRNAs in CSF and serum of patients within the acute period (1 and 5 days) of SCI (Appendix A). The authors were able to partially confirm the data obtained previously using the model of pig SCI. Thus, an increase in the concentration of miRNAs in CSF, dependent on the severity of SCI, was noted in the first 24 h after neurotrauma, followed by a decrease in in CSF miRNA levels by 3 days to similar abundances as those observed in non-SCI control patients. The authors report that a number of miRNAs that are differentially expressed in pig SCI are also altered during the acute phase of human SCI. MiR-208b-3p and miR-499 demonstrate similar expression patterns in serum samples from both the pig model and patients and are associated with severity of damage. Fifty percent of miRNAs, which were differentially expressed depending on severity in porcine serum, were also significantly altered in humans after SCI. In patients with SCI, the greatest changes in the levels of miRNAs (at least 190) were detected in the CSF, whereas in contrast, the experiments in pigs showed a stronger association of miRNAs detected in serum with severity of injury and neurological outcome [19].

Despite the work carried out and the detected changes in the miRNA profile both at the local and systemic levels in SCI, the pathophysiological significance of miRNA has yet to be determined. The data obtained in the abovementioned works show that changes in miRNA expression may contribute to the pathogenesis of SCI, making miRNAs not only potential targets for therapeutic interventions but also potentially effective biomarkers of severity of disease [27,36].

### 2.2. Neurodegenerative Diseases Exemplified by Alzheimer’s and Parkinson’s Disease

Currently, the most actively developed area for the study of biomarkers in the neurodegenerative field is for Alzheimer’s disease (AD) [2,37]. Several studies show that miRNA dysregulation may be associated with neurodegeneration in AD [27]. More than 300 miRNAs were found in the study of the hippocampus, medial frontal gyrus and cerebellum in the early and late stages of AD. Dysregulated miRNAs have been linked to new and known molecular pathways in AD, such as neurogenesis, oxidative stress, insulin resistance and innate immunity [37].

In a study by Denk et al. (2015), 74 miRNAs were downregulated and 74 were upregulated in the CSF of patients with mild late-onset AD and mild cognitive impairment. Significant differences, when compared with a non-AD control group, were confirmed for 12 of these miRNAs, the targets of which are involved in the regulation of tau and amyloid pathways in AD, such as MAPT, BACE1, and mTOR. Using discriminatory analysis with a combination of three miRNAs (miR-100, miR-103 and miR-375), the possibility of detecting AD in CSF was established with an accuracy of ~96% [38]. Subsequently, the authors studied the profile of circulating miRNAs from CSF and blood serum in frontotemporal lobar degeneration and AD (diagnosed according to criteria from the National Institute of Neurological and Communicative Diseases and Stroke (NINCDS) and the Alzheimer’s Disease and Related Disorders Association (ADRDA)) and compared to cognitively healthy control cases [39]. The authors found that circulating miRNA levels in CSF are lower and do not correlate with serum miRNA levels. A decrease in miR-320a expression was found, as well as an increase in miR-26b-5p, miR-30d-5p and miR-30b-5p in patients with AD and frontotemporal lobar degeneration compared to controls. A correlation between miR-30b-5p and the density of amyloid plaques had already been established [40].

Studies by van Harten et al. (2015) compared the CSF miRNA profiles of AD patients who met the NINCDS-ADRDA and NIA-AA (National Institute on Aging and the Alzheimer’s Association) criteria with high probability of AD etiology based on CSF biomarkers. There was decreased expression of miR-let-7a and miR-532-3p in the subgroup of early-onset AD patients. It had already been reported that miR-let-7a expression is reduced in the white matter but not in the cerebral cortex of AD patients [41]. It has been suggested that the decrease in miR-let-7a expression may be functionally related to the expression of AβPP and neurodegeneration processes in AD [42]. In a similar study of CSF from AD patients with similar clinical manifestations, Dangla-Valls et al. (2017) noted increased expression of both miR-125b and miR-222, which, as previously shown, induce phosphorylation of the tau protein by increasing expression of several tau kinases and inhibiting expression of phosphatases [43,44]. In addition, there is evidence that miR-125b may be associated with astrogliosis in neurodegeneration [45]. It has been hypothesized that miR-222 promotes increased expression of matrix metalloproteinases (MMPs) by repressing their inhibitor TIMP3, which may lead to increased neuronal apoptosis and inflammatory processes that promote neurodegeneration [34]. Marchegiani et al. (2019) investigated the expression of 4 miRNAs, including miR-125b and miR-222, in the CSF of patients with AD and other tauopathies, vascular dementia and cognitively normal subjects to identify diagnostic and prognostic markers of dementia. Of the abovementioned miRNAs, only miR-222 was significantly increased in patients with vascular dementia compared to the rest of the study groups. The previously suggested diagnostic significance of miR-21, miR-146a and miR-125b CSF expression levels has not been confirmed [46].

In a study by Riancho et al. (2017), changes in the expression of 15 miRNAs from exosome-enriched CSF samples from patients with a clinical presentation consistent with AD were analyzed. Of these 15, three miRNAs were selected as potential biomarkers of AD: miR-9-5p, miR-134 and miR-598. Decreased expression of miR-9-5p and miR-598 in CSF of AD patients confirmed the hypothesis that these miRNAs may be downregulated in response to neurodegenerative processes. Additional analysis using the miRSystem showed that these miRNAs are able to regulate gene pathways associated with amyloid proteins, stress pathways and neurotrophic signaling, which confirms a potential role in AD pathogenesis [47]. Interestingly, these data are consistent with the results of earlier studies, which described a decrease in expression of both miR-9-5p and miR-598 of CSF in AD [40,48].

Clinical diagnosis of both AD and Parkinson’s disease (PD) is difficult in the early stages of the disease. Reliable biomarkers are needed to diagnose these neurodegenerative diseases and track their progression. Since AD and PD belong to the same group of diseases and have similar difficulties in diagnosing the pre-symptomatic stage, comparative studies of the transcriptome profile of the CSF of such patients are often carried out. We believe that CSF is a more reliable source of biomarkers. CSF has the advantage of being in direct contact with the CNS and reflecting a more stable brain signature due to its proximity to the diseased tissue. Additionally, data from Burgos et al. show that CSF microRNAs are capable of reflecting cellular changes in brain tissues. Serum is a less invasive source of biomarkers [40]. However, microRNAs from peripheral blood are signals from all human organs and tissues, which makes their analysis more difficult. In general, there was a difference in expression of miRNA in postmortem samples of CSF and blood serum between patients with AD and PD (with an average duration of the disease of 7.5 ± 4.1 and 12.6 ± 7.9 years, respectively) when compared with the control group [40]. The authors determined a decrease in miR-9 and miR-101 in CSF in patients with AD and PD compared to the control group, which is confirmed by earlier studies [49,50]. It is assumed that the suppression of miR-101 may significantly contribute to the pathology of AD through signaling pathways associated with *APP, NFT* and *COX-2*.

Dos Santos et al. (2018) studied exosomal miRNAs from the CSF of patients with akinetic-rigid subtype PD, an average disease duration of ~2 years and UPDRS III 6, 29 (Unified Parkinson’s disease rating scale III). It was found that PD patients have higher levels of expression of miR-let-7f-5p and miR-151a-3p, as well as lower levels of expression of miR-27a-3p, miR-125a-5p and miR-423-5p, when compared to control groups. The authors identified 31 signaling pathways that are regulated by the above miRNAs and are involved in the pathogenesis of PD. In addition, the authors proposed a model for detection of PD in the early stages, which indicates patients with PD will have low levels of α-syn protein and decreased expression of miR-22-3p, as well as high levels of expression of miR-10b-5p and miR-151a-3p, in their CSF. Experimental confirmation of the validity of the proposed model has yet to be carried out, especially since earlier studies have shown a decrease in miR-151 expression in patients with PD but in those with a longer course of the disease [51,52]. Three additional miRNAs have been proposed as CSF biomarkers. An increase in miR7-5p and miR-331-5p, as well as a decrease in miR-145-5p, have been linked with good diagnostic accuracy in PD, with an average disease duration of ~7 years [53]. Gui et al. (2015) also reported an increase in the expression of miR-331-5p in the CSF of PD patients with the disease for more than 10 years.

A gradual decrease in miR-132 expression was found in the CSF of PD patients as the disease progressed compared to the control group. Expression of miR-132 was previously described as essential for morphogenesis and neuronal function, while significant suppression of miR-132 expression was associated with impaired neuronal function [54]. One of the targets of miR-132 is the CpG-binding protein, which is an important component of the development of the nervous system and neurodegeneration; therefore, miR-132 is a promising marker for diagnosis and treatment of PD [40].

Together, these studies are a first step towards assessing the expression of miRNAs in CSF and their changes in the course of AD and PD. Despite the early stage of this research, studies are already emerging on the use of miRNA mimics to interfere with target genes, which may have potential in the treatment of PD and AD [55,56,57]. However, the specificity of miRNAs and the reproducibility of their effects have yet to be confirmed. Moreover, these studies cover a fairly wide range of disease course and a limited sample of patients. Therefore, in order to use miRNAs as reliable biomarkers of neurodegenerative diseases, it is necessary to create not only standard working protocols for collecting, storing and analyzing samples but also to form a patient database with comprehensive information on duration and severity of the disease.

## 3. Methods for Isolation of miRNAs from Cerebrospinal Fluid

Quantification of circulating miRNAs is difficult due to their low concentration, effects of cell contamination and the absence of endogenous elements for normalization [27]. Body fluids, such as blood serum, plasma and CSF, contain low concentrations of total RNA, of which miRNAs constitute only a small part. In addition, the study of the miRNA profile in CSF is hampered by the small volume of samples obtained [58]. Despite the complexities of RNA isolation from CSF described above, there has been significant effort in the development of more efficient methods of RNA isolation.

In a study by Burgos et al. (2013), the efficiency of RNA isolation using nine commercial kits was tested, determining the concentration of the total RNA obtained from serum, plasma and CSF samples from a person with an unspecified diagnosis. CSF samples were prepared in 500 to 1500 μL aliquots, to which a known amount of synthetic exogenous small RNAs (referred to as a spike-in) was added as a positive control. In all the test kits, mixtures of guanidinium-thiocyanate and/or phenol-chloroform were used, allowing the sample to be separated into an aqueous phase with RNA and an organic phase containing proteins and DNA. For the extraction of RNA from the aqueous phase, some of the kits (Max Recovery Bioopure RNA, TRI Reagent RT, TRI RT Blood, TRI Reagent RT—Liquid Samples and RNAzol) were based on the precipitation of RNA with ethanol (or isopropanol), while others (mirVana, mirVana Paris and miRNeasy) used solid-phase extraction and subsequent adsorption on a fiberglass membrane. It is assumed that the use of kits for the extraction of RNA in the aqueous phase with subsequent precipitation allows a higher yield of RNA to be obtained compared to columnar kits for extraction. Since it is believed that RNA from the filters on the columns may not be completely washed out, losses in the total amount of total RNA are possible. Despite these assumptions, three of the four most effective kits were based on columns with a membrane filter for RNA isolation. RNA extraction kits demonstrated a yield of total RNA from CSF of 15 to 30 ng per mL. In addition, it was shown that the concentration of RNA varied among the technical repeats using all of the kits. Despite the fact that the authors identified the most efficient kits for RNA isolation from CSF (MaxRecovery BiooPure RNA Isolation Reagent, mirVana miRNA Isolation Kit, mirVana PARIS), the amount of RNA obtained was too low for subsequent sequencing.

Considering the above, Burgos et al. (2013) further optimized RNA extraction from plasma and CSF samples by rehydrating the interfacial and organic phases and re-extracting the RNA. It was found that with repeated extraction with phenol-chloroform, it is possible to increase the yield of miRNA by almost two times. However, repeating this step a third or fourth time did not lead to a significant increase in the amount of miRNA isolated. A volume of 500 μL of CSF was found to be sufficient for the extraction of RNA required to obtain reproducible miRNA analysis results [58]. Subsequent studies have sought to further enhance the yield of recovered RNA through the use of glycogen or exogenous RNA (east + RNA or MS2 phage RNA) to act as a carrier to increase the RNA yield at the isolation stage [59,60]. The results obtained were somewhat ambiguous, as the increase in the concentration of RNA after the addition of glycogen only occurred with some of the extraction kits [59].

In Akers et al. (2017) the authors investigated extracellular vesicle (EV) -derived miRNA from tumor tissue and CSF of patients with glioblastoma. The EV fraction from the CSF was isolated by a differential centrifugation method. EV pellets were resuspended in PBS; then, miRNA was isolated using the miRCURY RNA Isolation Kit. The amount of obtained miRNAs was estimated using real-time PCR since the concentration of the isolated RNA did not exceed 20 ng/μL. In addition to EV-derived miRNA, the authors examined miRNA from whole CSF and found that more miRNA species were found in whole CSF compared to the EV-enriched fraction. The authors also identified, for the first time, nine miRNAs from CSF as potential biomarkers of glioblastoma [61]. A similar study was carried out by Kopkova et al. (2018a) [62], who compared several approaches for the isolation of miRNA from CSF patients with glioblastoma and identified the most efficient miRNA purification kit as Norgen Biotek.

In a study by Gui et al. (2015) EV-derived miRNA from the CSF of patients with AD and PD obtained by differential centrifugation were extracted using the Qiagen miRNeasy Serum/Plasma Kit according to the manufacturer’s instructions. The authors tried to profile the expression of 746 miRNAs using TaqMan arrays. However, only 132 miRNAs (17.7%) were found, of which 27 and 6 miRNAs were differentially expressed in PD and AD, respectively, compared to the control group [52].

Circulating and EV-derived miRNAs in the CSF (1 mL samples) of patients with various neurological disorders were investigated by Saugstad et al. (2017). For comparative analysis of the efficiency of isolation of total and EV-derived miRNA, four commercial kits were used. As expected, the RNA yield was higher when using kits specific to the isolation of total miRNA. The greatest difference was observed in CSF samples from patients with glioblastoma multiforme, where the total RNA obtained using the mirVana and miRCURY kits was in the range of 1.5–3.2 ng/μL, compared to 0.00–2.22 ng/μL of total RNA isolated using exoRNeasy and Total Exosome [63]. 

To determine biomarkers in acute ischemic stroke, next-generation sequencing was performed using miRNAs isolated from 100 μL CSF using TRIzol reagent [64] and compared to real-time PCR analysis of miRNAs isolated from 200 μL of CSF from a similar sample set using the miRCURY RNA Isolation kit for biofluids or the Norgen Biotek Total RNA Purification kit. The two analysis platforms, sequencing and real-time PCR, did not lead to identical results. The authors considered the reasons for this to be the relatively small sample set and a heterogeneous control group. Patients in the control group had different diagnoses, which may have influenced the changes calculated in relation to the group of people with stroke. The authors were also unable to confirm the results of their earlier pilot study [65], highlighting the importance of further research, especially with a small sample of patients.

Obviously, obtaining high-quality RNA is the first step in studying miRNAs in CSF. The stage of RNA isolation from CSF plays an important role because it determines how complete the understanding of the miRNA profile will be. Since the concentration of miRNA in CSF is low, its isolation is difficult. Despite the fact that a fairly large number of different commercial kits for the isolation of RNA from various tissues and biological fluids have been developed, researchers are constantly looking for new protocols with good reproducibility and efficiency. Until now, there is no single standard, well-proven protocol for the isolation of RNA from CSF, which can lead to low interlaboratory reproducibility of the results and the objectivity of the data obtained.

## 4. Assessment of the Quality of the Obtained miRNA

Analysis of the obtained miRNA isolated from CSF is most often performed using the Bioanalyzer capillary electrophoresis system (Figure 1). This system has a high sensitivity of 50 pg/μL for RNA and requires minimal sample volumes (1 μL) to obtain accurate results [66]. In addition, the system requires a minimum number of manipulations from the operator and is economical in terms of analysis time (25–45 min), depending on the number of samples and the chip used [67]. However, the Bioanalyzer only shows the size of the RNA but does not identify RNA fragments, which may be disrupted rRNA, mRNA or long non-coding RNA. It is also worth noting the high cost of the device and reagents, which have a short shelf life, making this type of analysis less accessible.

Evaluation of the isolated miRNA is also possible using spectrophotometers, such as the NanoDrop. However, the NanoDrop is rather limited in its ability to detect the low levels of miRNA obtained from CSF. Our experience has shown that NanoDrop measurement accuracy depends on the frequency of instrument use/calibration and the nature of the samples being analyzed. A common alternative to the NanoDrop is the Qubit fluorometer. It works by adding a fluorescent probe to the sample, which specifically binds the target molecule to form a fluorescent complex. The light source produces an intensity of fluorescence that is proportional to the concentration of the analyte. The device measures the intensity of the glow and calculates the concentration. Using a fluorometer, the concentration of RNA can be quickly and accurately determined, while new models of the device allow for measurement of the amount of miRNA. However, the sensitivity of the instrument is low and the range of quantitation starts at 5 ng. Use of a spectrophotometer, a fluorimeter and microfluidic electrophoresis can provide information on the integrity and the amount of miRNAs, but unfortunately, there is a possibility of obtaining data with significant variability [38,42,48,63].

The low sensitivity of the above systems for assessing the quantity and quality of miRNA from CSF has led to the use of other microfluidic technologies, such as TaqMan Low-Density Array (TLDA), based on real-time PCR (Figure 1). TLDA uses single-plex PCR with mapped primers for analysis. In this case, it is possible to add your own primers and probes, as well as to use a ready-made calibration curve. The advantage of this system is the speed and convenience of sample preparation, since reagents for each assay are already applied to the well [68]. The disadvantage of this system is the inability to identify and measure new types of miRNAs and the average productivity by the criterion of the number of samples analyzed per day [67].

The use of qRT-PCR is an alternative method. When used with a known amount of control miRNAs, it is possible to establish the efficiency of RNA extraction [67]. This method is highly reproducible, fast and easy to use [69]. However, existing commercial qRT-PCR kits use a strategy based on reverse transcription of mature miRNA molecules, which can lead to various detection errors. Another obstacle to qRT-PCR is that reaction conditions can vary and miRNA primers are difficult to design [70]. The latter is associated with the length of mature miRNAs, which is insufficient for annealing with traditional primers designed for reverse transcription and PCR. In addition, miRNAs do not have a common sequence, for example, as poly (A) tail, which can be used for enrichment or as a universal primer-binding site for reverse transcription [67]. 

Of course, the analysis of miRNA obtained after RNA extraction from CSF is very important since inaccurate assessment can greatly affect the diagnostic ability of these molecules [69]. Therefore, further development of the most efficient and reproducible method for quantitative and qualitative assessment of the obtained miRNA is required.

## 5. Methods for Studying the miRNA Expression Profile

There are several approaches to studying miRNA expression. It is possible to study the expression of one or several specific miRNAs or perform total genome-wide profiling of miRNAs. To date, the expression of specific types of miRNA in CSF is measured using real-time PCR [69]. The most common method is specific reverse transcription with stem-loop primers and measurement of real-time PCR expression using a probe-based detection system, such as TaqMan [38,43,58,71,72]. Another method for measuring miRNA expression is based on universal reverse transcription, followed by SYBR Green quantitative PCR with specific forward and reverse primers [47,60]. Digital PCR can also be used with the TaqMan chemical protocol to analyze the level of miRNA expression [62,73]. Digital PCR data were confirmed using next-generation sequencing analysis [73]. The above technologies based on qRT-PCR are effective methods for determining the level of miRNA expression in samples with low RNA content, such as CSF and serum. Real-time PCR is also used as the main method for validating sequencing- and microarray-based analyses discussed below [67,74].

One of the first methods for simultaneous analysis of a large number of specific miRNAs was based on hybridization with microarrays. The advantage of miRNA microarrays lies in their low cost and in the possibility of simultaneous analysis of a large number of samples. The disadvantages of this method are the limited range of quantification, imperfect specificity for miRNAs that differ from each other by one nucleotide and the inability to perform an absolute quantification of the miRNA profile. Thus, the optimal field of application of microarray technology is considered to be in determination of the relative amounts of specific miRNAs between two different states [67].

The NanoString nCounter multiplex system uses oligonucleotide tags for barcoding, followed by target detection by hybridization with color-coded probes. The system can quantify up to 800 different targets. The advantages of the NanoString nCounter system are the direct measurement of the expression levels of molecules without amplification and the ability to investigate the expression of formalin-fixed and paraffin-embedded samples [75]. One of the main advantages of this method is the ability to discriminate miRNAs differing in one nucleotide with high accuracy. However, data analysis using the NanoString nCounter system is more expensive than using existing counterparts [67,76].

For total profiling of miRNA expression, next-generation RNA-seq sequencing technology is used. The initial stage is the preparation of a cDNA library from small RNAs of the test sample, followed by sequencing. Bioinformatic analysis of sequence reads identifies both known and new miRNAs within the sample and provides relative quantification. The main advantages of next-generation sequencing for miRNA profiling are the ability to identify new miRNAs and the precise differentiation of miRNAs that differ by one nucleotide [77]. Limitations of next-generation sequencing can be the high cost and limited amount of “barcoding” cDNA, which does not allow for the loading of multiple samples in one run [67]. Sorensen et al. (2017) concluded that the study of miRNA expression in CSF using RNA-seq is less sensitive when compared with qPCR. At the same time, Tigchelaar et al. (2019) confirmed the relationship between the severity of SCI and the level of miR-10b-5p expression in CSF using both NGS RNA-seq and qPCR [19,64].

The choice of a method for studying the profile and expression of miRNA depends on the quantity and quality of the isolated RNA. Chips for hybridization, technologies based on real-time PCR, the nCounter platform and next-generation sequencing differ in the concentration and volume of the miRNA sample required for analysis, as well as in sensitivity, specificity and cost. Whilst all aspects of profiling should be considered before a platform [69], we propose that the use of miRNA-Seq technology to detect miRNAs and their subsequent validation using qPCR can act as effective method for establishing reliable biomarkers of SCI and neurodegeneration diseases.

## 6. Methods for Control and Normalization of miRNA Expression

Various factors affect the levels of miRNAs in biological fluids, for example, methods of material collection, the method of isolation of the miRNAs and the amount of miRNAs in the sample. Therefore, in order to efficiently analyze miRNAs in CSF, it is necessary to standardize their selection and isolation conditions and use exogenous and endogenous control miRNAs to normalize the expression of data [78]. The use of exogenous miRNA, which is absent in the studied organism (for example, miRNA from *Caenorhabditis elegans* or plants), makes it possible to determine technological variability [40,59,79]. It was previously established that it is necessary to add exogenous RNA directly to the sample with CSF and lysis solution since endogenous RNases can destroy it immediately [69,80]. 

To demonstrate possible changes in biological factors before RNA is isolated from CSF, it is possible to use stable endogenous miRNAs or other short, noncoding RNAs that do not change against the background of the studied pathological conditions [69]. Endogenous controls for CSF have been previously defined [52,64,81]. However, choosing the correct endogenous control is not an easy task since no miRNA molecule shows stable expression in CSF under various pathological conditions [82].

For RNA sequencing, the next stage of normalization occurs during the primary bioinformatics processing of the data, where the detection of the RNAs is expressed as counts per million. During analysis, reads are compared with a specific miRNA, divided by the total number of aligned reads and multiplied by one million [19]. Another approach for counting miRNAs in CSF was proposed by Sørensen et al. (2017). The authors used a truncated mean M-value normalization method to estimate relative miRNA levels from a RNA-seq data. 

In addition to quantitative and qualitative analysis of miRNAs, it is necessary to determine their potential targets and effects. MiRBase is the major online database for all miRNA sequences and annotations, allowing for the prediction of gene targets [83,84]. The latest version of miRBase (v22) contains information on miRNAs from 271 organisms: 38,589 hairpin precursors and 48,860 mature miRNAs. However, miRBase does not provide an analysis of the functions of miRNAs, and the articles to which links are given usually provide information about the time and source of the discovery of miRNAs but not about their function. miRBase does not curate or match predicted or verified target sets but, rather, links records to external target resources [85]. Information on miRNA functions can be obtained from other databases, such as miRTarBase, TargetScan, DIANA-microT and miRDB. Using miRTarBase, one can obtain information about miRNA targets that have experimental confirmation [86,87,88].

To summarize, an important step in profiling miRNAs in CSF is control and normalization of the data obtained. This is a complex and important question that needs to be addressed in the study of circulating miRNAs. Since the capabilities of devices are limited and do not allow for an accurate assessment of the quantity and quality of miRNA from CSF, researchers are forced to introduce additional control steps to objectively assess the expression of miRNA. Bioinformatic analysis is an integral part of sequencing, which helps not only to establish the miRNA profile but also the number of copies in the studied samples. Special databases are usually used to determine the targets and possible biological effects of a particular miRNA. However, to confirm the participation of miRNAs in a particular process, additional, more in-depth studies must be carried out.

## 7. Conclusions

At the present time, there are many studies aimed at elucidating the molecular mechanisms of the course of different neurological disorders. The issues of early diagnosis of neurodegenerative diseases and the prognostic capabilities of biomarkers in SCI still remain. Researchers are searching for biomarkers not only in affected tissues and organs but also in biological fluids for the use in minimally invasive diagnostics in the future. The most successful source of biomarkers for neurological disorders can be considered to be within the CSF because of it is in direct contact with the brain. However, it should be borne in mind that the collection of CSF is an invasive procedure for which there are contraindications that need to be considered.

As biomarkers, it is proposed to consider miRNAs that are involved in the regulation of gene expression and therefore involved in the abovementioned pathological processes. Nevertheless, the study of the miRNA profile in CSF is a rather difficult task associated with (1) a low concentration of miRNA in CSF; (2) the need to introduce controls at each stage; (3) the variability of the data obtained to assess the quantity and quality of isolated miRNA, associated with the technical features of various devices; (4) establishing a reliable picture of gene regulation and the influence of miRNA on specific mechanisms of pathogenesis; and (5) low interlaboratory reproducibility of results and objectivity of the data obtained. Considering the above, it should be noted that before using miRNAs as diagnostic and prognostic agents, it is necessary to establish unified approaches for the isolation and analysis of the miRNA profile of human CSF, with clear standards for clinical use.

## Figures and Tables

**Figure 1 ijms-23-00114-f001:**
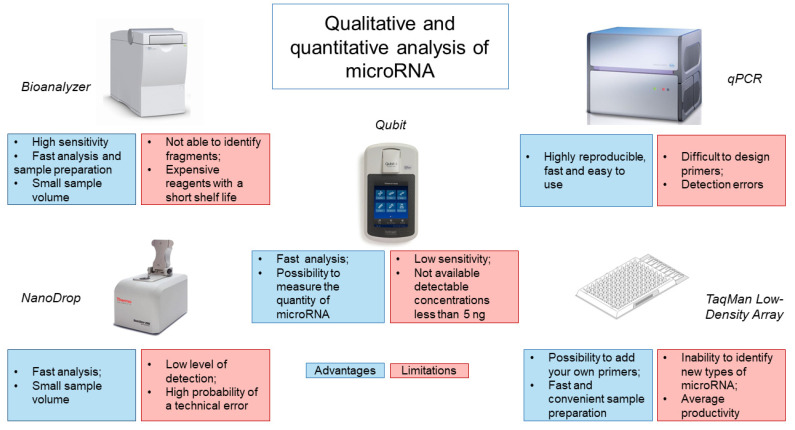
Analysis of methods for determining the quantity and quality of microRNA from CSF.

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
