# Peer review of "The Study of Cerebrospinal Fluid microRNAs in Spinal Cord Injury and Neurodegenerative Diseases: Methodological Problems and Possible Solutions"

_ijms, 2021, doi:10.3390/ijms23010114_

Round 1

Reviewer 1 Report

This is a very useful and timely review article concerning mRNAs in CSF in neurological diseases. A summary table of mRNAs observed in CSFs in the various diseases would be helpful. 

Line 206 Maybe this should read: with AD, and other tauopathies

Line 377 intensity of the fluorescence 

In the final conclusions, perhaps consideration should be given about the risks to the patient of CSF sampling.

Author Response

Reviewer #1:
1. This is a very useful and timely review article concerning mRNAs in CSF in neurological diseases. A summary table of mRNAs observed in CSFs in the various diseases would be helpful.

Authors: We would like to thank the reviewer for their consideration and support of our manuscript. Our review is devoted to a detailed examination of microRNAs in the cerebrospinal fluid and the possibility of using these molecules as biomarkers. We would like to include the table for mRNA in our future work, which will directly concern genes in various disorders of the central nervous system.

  1. Line 206 Maybe this should read: with AD, and other tauopathies

Authors: According to the reviewer’s comment, we have changed line 206 with AD, and other tauopathies.

  1. Line 377 intensity of the fluorescence

Authors: According to the reviewer’s comment, we have changed line 377 intensity of the fluorescence.

  1. In the final conclusions, perhaps consideration should be given about the risks to the patient of CSF sampling.
    Authors: According to the reviewer’s comment, we have added into the final conclusions information about the risks to the patient of CSF sampling.

We would like to thanks the reviewer for reviewing and pointing out our errors. As it becomes clear from our responses to specific comments above, we have tried to improve the presentation of our article as much as possible.

Reviewer 2 Report

This is a comprehensively written review focusing the use of CSF miRNA. This also highlight the current technical difficulties in the field. It could me more interesting if the authors can compare between Serum and CSF miRNA to demonstrate why CSF miRNA is a more reliable marker than serum miRNA. This is briefly discussed in line 226-230. Expanding this will be more useful.

miRNA after SCI in pigs were referenced in the review (line 135-143). I tis mentioned that there is a significant change in miRNA. Authors are requested to explicitly report whether there is any increase or decrease of specific miRNAs.

Author Response

Reviewer #2:
1. This is a comprehensively written review focusing the use of CSF miRNA. This also highlight the current technical difficulties in the field. It could me more interesting if the authors can compare between Serum and CSF miRNA to demonstrate why CSF miRNA is a more reliable marker than serum miRNA. This is briefly discussed in line 226-230. Expanding this will be more useful.

Authors: We would like to thank the reviewer for their consideration and support of our manuscript. We have added several proposals on the possibilities of using CSF and serum as sources of microRNA.
2. miRNA after SCI in pigs were referenced in the review (line 135-143). I tis mentioned that there is a significant change in miRNA. Authors are requested to explicitly report whether there is any increase or decrease of specific miRNAs.

Authors: We decided to remove the sentence «The severity of SCI had a significant effect on the levels of miRNAs in serum, in which there was a significant change in the expression of 58, 21 and 9 miRNAs between severe, moderate and mild SCI.» and we have added specific microRNAs that have changed after SCI.

We would like to thanks the reviewer for reviewing and pointing out our errors. As it becomes clear from our responses to specific comments above, we have tried to improve the presentation of our article as much as possible.